# Clues to tRNA Evolution from the Distribution of Class II tRNAs and Serine Codons in the Genetic Code

**DOI:** 10.3390/life6010010

**Published:** 2016-02-24

**Authors:** Harold S. Bernhardt

**Affiliations:** Department of Biochemistry, University of Otago, Dunedin 9016, New Zealand; harold.bernhardt@otago.ac.nz; Tel.: +643-479-7841

**Keywords:** transfer RNA, tRNA evolution, genetic code, small hydrophilic amino acids, class II tRNAs, tRNA introns, aspartic protease, recycling

## Abstract

We have previously proposed that tRNA^Gly^ was the first tRNA and glycine was the first amino acid incorporated into the genetic code. The next two amino acids incorporated would have been the other two small hydrophilic amino acids serine and aspartic acid, which occurred through the duplication of the tRNA^Gly^ sequence, followed by mutation of its anticodon by single C to U transition mutations, possibly through spontaneous deamination. Interestingly, however, tRNA^Ser^ has a different structure than most other tRNAs, possessing a long variable arm; because of this tRNA^Ser^ is classified as a class II tRNA. Also, serine codons are found not only in the bottom right-hand corner of the genetic code table next to those for glycine and aspartic acid, but also in the top row of the table, next to those for two of the most hydrophobic amino acids, leucine and phenylalanine. In the following, I propose that the class II tRNA structure of tRNA^Ser^ and the arrangement of serine codons in the genetic code provide clues to the early evolution of tRNA and the genetic code. In addition, I address Di Giulio’s recent criticism of our proposal that tRNA^Gly^ was the first tRNA, and discuss how early peptides produced from a restricted amino acid alphabet of glycine, serine and aspartic acid might have possessed proteolytic activity, which is possibly important for the early recycling of amino acid monomers.

## 1. Introduction

We have previously proposed that tRNA^Gly^ was the first tRNA, and glycine was the first amino acid incorporated into the genetic code, with the universally-conserved glycine codons GGN the first codons to be assigned due to their strong base-pairing interaction with the tRNA^Gly^ anticodon loop sequence (NCC)A (anticodon in brackets; N is any of the four bases) [1,2]. The early incorporation of glycine into the genetic code is supported by the ease of its synthesis under plausibly prebiotic conditions [3,4]. Duplication of the tRNA^Gly^ sequence followed by mutation of its NCC anticodon by a single C to U transition mutation—possibly through spontaneous deamination—would have produced tRNAs with either NCU or NUC anticodons [5]. While these mutant tRNAs would initially still have coded for glycine, subsequent mutations in the “operational code” contained in their acceptor stems [6] would have altered their aminoacylation specificity to serine (for NCU anticodons) and aspartic acid (for NUC anticodons). Although in the contemporary code serine shares the AGN codon box with arginine (Figure 1), the latter was probably incorporated into the genetic code at a later stage, as it is not produced in early earth simulation experiments [7] and has a complex biosynthetic pathway; it is possible that the arginine codons in the AGN codon box are the result of more recent anticodon “creep” by tRNAs^Arg^ with anticodons complementary to codons in the adjacent CGN codon box (Figure 1). Also, aspartic acid shares the GAN codon box with glutamic acid in the modern code; the close functional similarity between the two amino acids suggests that originally the code may not have discriminated between them, with the GAN box only formally divided at a later stage ([8]; and see below for further discussion of this point). 

It should be noted that the hypothesis presented here addresses early tRNA evolution in the context of an RNA world in which tRNA sequences were able to be reliably replicated, albeit with a (significantly?) higher error rate than exists in modern systems. As the proposed steps represent the very first assignments of the genetic code, they would necessarily have occurred prior to the advent of coded protein synthesis and therefore before the production of complex catalytic proteins such as the protein aminoacyl-tRNA synthetase enzymes. At the stage which the hypothesis addresses, tRNA aminoacylation would have been carried out by aminoacylating ribozymes, a scenario supported by the demonstration by Yarus and colleagues that even extremely small ribozymes are able to catalyze the aminoacylation of RNA [9].

## 2. Hypothesis

I propose three stages in the evolution of class II tRNAs:
Evolution of tRNA^Ser^(NCU). During the initial duplication and anticodon mutation of tRNA^Gly^(NCC) to form tRNA^Ser^(NCU), an enlargement of the variable arm (V arm) occurred—possibly through error-prone replication or retention of an intron located in the V arm region [10]—which produced the long V arm characteristic of a class II tRNA structure (Figure 2).Evolution of tRNA^Ser^(NGA). Some time following its appearance—and possibly much later—the NCU anticodon of tRNA^Ser^(NCU) underwent two transversion mutations to produce tRNA^Ser^(NGA) (Figure 3, green arrows). Apart from having a different anticodon, tRNA^Ser^(NGA) retained a long V arm and the identity determinants that enabled aminoacylation with serine.Evolution of other class II tRNAs. Duplication of tRNA^Ser^(NGA) followed by various mutations of its anticodon gave rise to the other class II tRNAs which have codons in the codon boxes surrounding those of tRNA^Ser^(NGA): tRNA^Leu^, tRNA^Sec^, and bacterial tRNA^Tyr^ (Figure 3, blue arrows). We have previously proposed that the direction of amino acid incorporation into the genetic code was from the small hydrophilic to the larger hydrophobic amino acids [5]. According to this scenario, this third stage could have occurred relatively late in genetic code evolution, when the synthesis of larger peptides/proteins drove the incorporation of large hydrophobic amino acids that afforded stabilization to the synthesized proteins through formation of a hydrophobic core [11,12].


## 3. Discussion

As observed by Sun and Caetano-Anollés [14], previous hypotheses regarding tRNA evolution and the chronology of amino acid incorporation into the genetic code have tended to overlook the significance of the long variable arm. However, there would appear to be a contradiction between these authors’ statement that “the variable region was the last structural addition to the molecular repertoire of evolving tRNA substructures” and their belief that class II tRNAs (with a long V arm) evolved *prior* to class I tRNAs, with the first amino acids incorporated into the code being selenocysteine, tyrosine, serine and leucine [15].

Building on the ideas of Di Giulio [16] and Dick and Schamel [17], we have previously proposed that the first tRNA (RNA^Gly^) arose by the duplication—ligation of an RNA hairpin half the length of tRNA, with the ligation possibly catalyzed by a self—splicing intron positioned between nucleotides 37 and 38 in the anticodon loop of the resultant tRNA, the point of ligation between the two hairpins [1,2]. The rationale for this proposal was twofold: (1) This position represents the “canonical” intron position, where the majority of protein-cleaved tRNA introns occur in both Eukaryotes and Archaea [18]; and (2) duplication of an RNA hairpin with a 3’ CCA terminus (the universally conserved site of tRNA aminoacylation) would produce a full-length tRNA with the upstream hairpin’s CCA in positions 35, 36 and 37 of the single-stranded anticodon loop, precisely where it occurs as part of the universally conserved tRNA^Gly^ anticodon loop sequence. However, a number of Archaeal tRNAs have introns at other positions, including in the V arm of what are normally class I tRNAs, giving the unspliced tRNA transcripts the appearance of class II tRNAs. For example, Kjems and colleagues have discovered an 18-nucleotide intron in the V arm of the class I tRNA^Gly^(CCC) in the extreme thermophile *Thermofilum pendens*, leading them to propose that class II tRNAs may originally have been derived from class I tRNAs that retained splicing-deficient introns [10]. Introns are also found in the V arm of two lysine tDNA transcripts [18]. Other evidence for the evolution of class II tRNAs from class I tRNAs has been found in the nematode *Caenorhabditis*
*brenneri*, which was recently discovered to have five tRNAs possessing the large V arm typical of class II tRNAs in combination with anticodons usually found in class I tRNA (for glycine, valine, proline, and arginine) that cluster among class I tRNAs in a neighbor-joining phylogenetic tree [19]. Although the mechanism by which these tRNAs arose has so far not been elucidated—and they do not appear to be functional *in vivo* [20]—it demonstrates that the long V arm of class II tRNAs has arisen on more than one occasion in evolution, at least in nematodes. This evidence notwithstanding, the particular clustering within the genetic code table of codons read by tRNAs with a class II structure suggests—to this author at least—that these tRNAs have a common origin, having “descended” from tRNA^Ser^(NCU) (Figure 3). In support of this view, a structure-based phylogenetic analysis by Sun and Caetano-Anollés [14] of tRNAs from *Drosophila melanogaster* has demonstrated that the two sets of tRNA^Ser^ form a monophyletic group, suggesting that they have a common origin; however, see comment in [21].

Following its initial appearance, the large V arm may have been under strong positive selection, as it would have made it easier for the first aminoacylating ribozymes to discriminate between tRNAs. For example, a long V arm could have served as an anti-determinant for aminoacylation with glycine, much as it prevents cross-aminoacylation by modern protein aminoacyl-tRNA synthetase enzymes specific for class I tRNAs [22]. Moreover, at a later stage of enzyme evolution it could have served as a positive determinant: for modern protein seryl-tRNA synthetase enzymes, the presence of the long V arm of tRNA^Ser^—as opposed to its sequence—serves as the major identity determinant [23,24].

The discovery by Yarus and colleagues of aminoacylation catalyzed by an active site composed of only three nucleotides [9] suggests that the RNA world may have been characterized by ubiquitous aminoacylation. That this activity may have been accompanied by a degree of substrate promiscuity is suggested by the fact that the catalytic site of the Yarus ribozyme is able to utilize both phenylalanine and methionine as substrates [25], possibly due to the similarity in size and hydrophobicity of the two amino acids. If such promiscuity—or lack of specificity—were a general property of aminoacylating ribozymes, this might be consistent with our hypothesis that the first amino acids incorporated into the genetic code were the small, hydrophilic amino acids, as their similarity may have made them all substrates for an aminoacylating ribozyme. In which case, tRNA^Gly^ may *not* have been the first tRNA, but rather, the tRNA that has retained the original NCC anticodon. Such a scenario would be consistent with the suggestion by Woese [8] that early genetic code assignments could have been ambiguous, *i.e.*, codons may have been assigned to *groups* of related amino acids rather than assigned to one amino acid in particular. In which case, Di Giulio may be correct in arguing that the first proto-mRNA did not only code for glycine [26], as it may also have coded for serine and aspartic acid. Interestingly, extant protein aminoacyl-tRNA synthetase enzymes also can exhibit a degree of substrate promiscuity, both for amino acids and tRNAs [27,28,29,30,31].

Fournier and Gogarten [10] have compared the relative amino acid usage between universally–conserved and domain–specific conserved positions in universally conserved ribosomal proteins and ATPase subunits, and uncovered strong evidence that glycine was the first amino acid incorporated into the genetic code, consistent with our proposal that tRNA^Gly^ was the first tRNA. Specifically, a strict consensus identified glycine, glutamine and leucine as “ancient” additions to the genetic code, and cysteine, phenylalanine, isoleucine, valine, tryptophan and tyrosine as “recent” additions, while a semi-strict consensus further identified proline as an ancient addition and glutamic acid, lysine and serine as recent additions, and the most permissive consensus additionally identified aspartic acid as an ancient addition [10]. As previously discussed, the hypothesis presented here concerning the evolution of the class II tRNA structure is consistent with our previous hypothesis [5] that the direction of amino acid incorporation into the genetic code was from the small hydrophilic to the larger hydrophobic amino acids, as leucine, tyrosine and selenocysteine are members of the latter group. However, an alternative hypothesis—represented by reversing the direction of the green arrows in Figure 3—is that the evolution of the two tRNAs^Ser^ occurred in the opposite direction, and that tRNA^Ser^(NCU) evolved from tRNA^Ser^(NGA). If this were correct, it would leave open the question of whether a tRNA^Ser^ was in fact the first class II tRNA to evolve (or even whether the class II tRNA structure might have arisen on more than one occasion). Although perhaps less likely, such a scenario would be consistent with Fournier and Gogarten’s finding that leucine was incorporated into the genetic code earlier than serine [10].

As noted above, mutation of the tRNA^Ser^(NCU) anticodon to NGA would have required two pyrimidine–to–purine transversion mutations: C to G, and U to A. Although rarer than transition mutations, transversion mutations of the tRNA anticodon still occur at a significant rate: 0.3–0.5 that of transition mutations [32]. In addition, it has been found that tRNA^Ser^(GGA) forms a particularly strong anticodon-anticodon interaction with tRNA^Gly^(U*CC) (U* is an unidentified derivative of U) [33], suggesting the possibility that this strong interaction might have driven fixation of tRNA^Ser^(NGA). It is interesting that the two sets of serine anticodons—NCU and NGA—are potentially complementary, although the significance (if any) of this is not clear. According to the Rodin-Ohno hypothesis, the genetic code originated through the coevolution of antiparallel RNA and β-hairpin peptide helices, with pairs of protein aminoacyl-tRNA synthetase enzymes from the two enzyme classes, and pairs of tRNAs possessing complementary anticodons, each originally coded for on complementary RNA strands [34]. If true, this potentially could offer an explanation for the anticodons of tRNA^Ser^(NCU) and tRNA^Ser^(NGA) being complementary.

According to the hypothesis presented here, tRNAs assigned to a number of hydrophobic amino acids with codons in the top row and left-most column of the genetic code table have a long V arm and are therefore derived from tRNA^Ser^(NGA) (Figure 3). However, not all of the amino acids with codons in this part of the table are coded for by class II tRNAs: tRNA^Phe^, tRNA^Cys^, and tRNA^Trp^ are all class I tRNAs, and so apparently have not evolved from tRNA^Ser^(NGA). Also, the fact that bacterial tRNA^Tyr^ has a class II tRNA structure in bacteria but not in archaea and eukaryotes suggests that the bacterial class II tRNA^Tyr^ is the ancestral version, and was subsequently replaced by a class I tRNA structure in the lineage leading to the archaeal and eukaryotic domains. As pointed out by Achsel and Gross [35], this replacement is an indication of the dynamic nature of the genetic code even after the divergence of the three kingdoms. A related dynamism is of course also seen in the variant genetic codes found in mitochondria and certain eukaryotic species [36].

Finally, we, along with others, have previously commented on possible activities of peptides composed of glycine and other small hydrophilic amino acids [5,37,38,39,40]. It is also possible that an important function that drove selection of the first, relatively simple, peptides was for their use as buffers (comment by Berthold Kastner in his published review in [2]). And prior to the evolution of protein synthesis, aminoacylation may have offered RNA a key functionality: a positively charged amino group not otherwise available to RNA at neutral pH (comment of anonymous reviewer). Regarding possible enzymatic functions of the first peptides, proteolysis—the ability to break peptide bonds and dissemble peptides into amino acids—is not perhaps what first comes to mind in terms of a selectable function. However, Cleaves has raised the possibility that “biochemistry has been selected for *recyclability* ([11]; italics in original)”, and suggested that proteolysis could have been a crucial activity of the first early peptides by maintaining the supply of amino acid monomers. (In this regard it is perhaps significant that the most ubiquitous catalytic activity of both naturally—occurring and *in vitro*—evolved ribozymes is for phosphoryl transfer reactions, which involve the cleavage and ligation of the RNA backbone). The catalytic site of aspartic proteases is formed by a pair of aspartic acid residues functioning as an acid/base pair, in the context of an aspartic acid-threonine-glycine-serine tetrapeptide sequence ([41]; with threonine replaced by serine in an ancestral peptide). Fournier and Gogarten have observed that incorporation of aromatic amino acids into the genetic code may have been important in the takeover of what had previously been functions of RNA nucleobases in an RNA world, such as hydrophobic stacking [10]. The obvious parallels between the hydroxyl groups of serine and ribose, and the acidic moieties of aspartate and phosphate groups suggest that the first amino acids incorporated into the genetic code could have replaced what were previously functions of the RNA ribose-phosphate backbone. However, more than simply replacing these RNA functions, proteins were both able to catalyze the majority of these reactions more efficiently (explaining the *nearly* complete replacement of ribozymes by protein enzymes; for an elegant exposition of this issue, see Poole *et al.* [42]) and to do what RNA could not: from the absence of reports of ribozymes capable of proteolysis, it appears likely that breaking peptide bonds is one reaction that is particularly difficult for RNA.

## Figures and Tables

**Figure 1 life-06-00010-f001:**
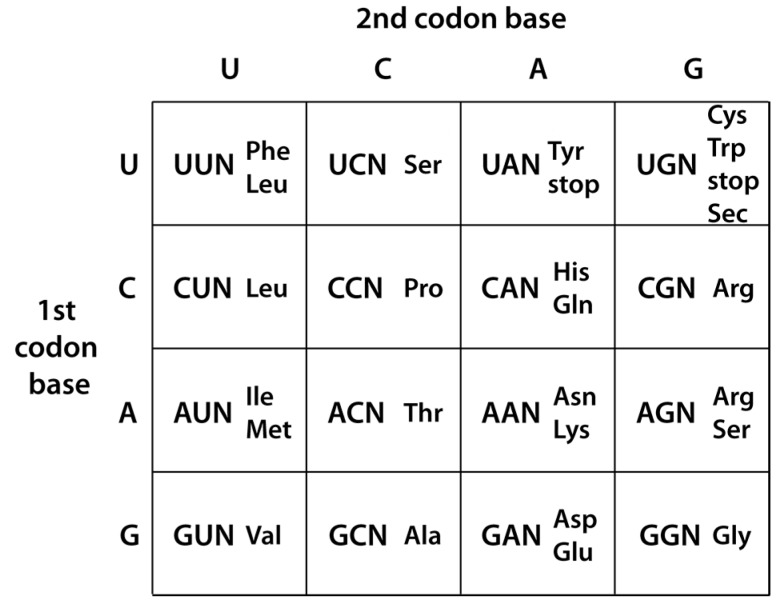
Genetic code table, in its standard representation. N is any of the four bases.

**Figure 2 life-06-00010-f002:**
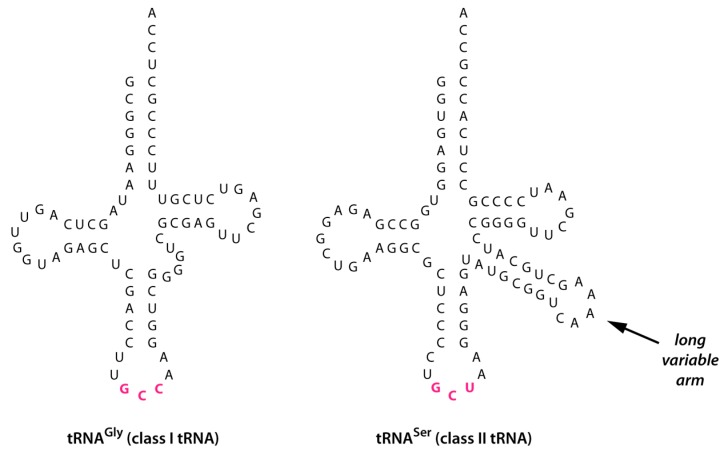
Secondary structures of class I and class II tRNAs, as represented by tRNA^Gly^(GCC) and tRNA^Ser^(GCU) from *Escherichia coli.* The tRNA sequences are shown without post-transcriptional modifications; anticodons are shown in pink. Sequences taken from [13].

**Figure 3 life-06-00010-f003:**
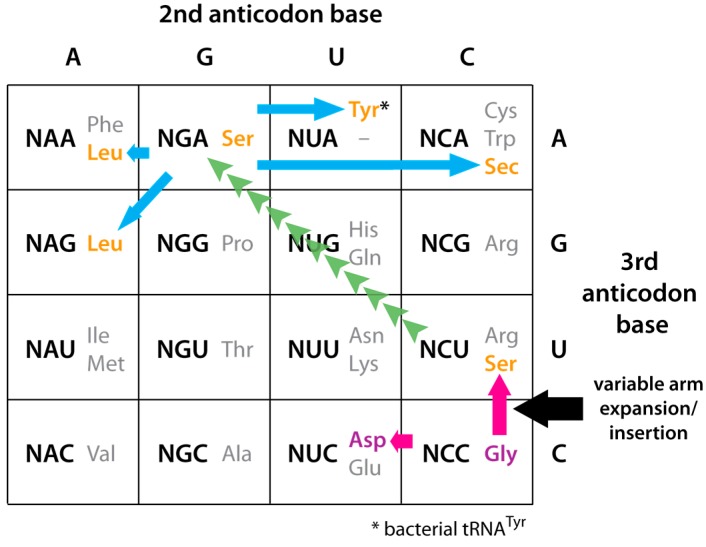
The genetic code table shown as tRNA anticodons, with the proposed direction of tRNA evolution indicated by colored arrows. 1. Evolution of tRNA^Asp^ and tRNA^Ser^ by single—possibly spontaneous—transition mutations (*pink arrows*), during which a loop insertion/expansion event in the variable arm of tRNA^Ser^ (*black arrow*) produced the class II tRNA structure. 2. Mutation of the NCU anticodon of tRNA^Ser^ by two transversion mutations to produce tRNA^Ser^(NGA) (*green arrows*). 3. Evolution of the other class II tRNAs (*blue arrows*). Amino acids corresponding to class I tRNAs are shown in purple; those corresponding to class II tRNAs are shown in orange. The remaining amino acids (*in grey*) are coded for by class I tRNAs.

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
