# Peer review of "Clues to tRNA Evolution from the Distribution of Class II tRNAs and Serine Codons in the Genetic Code"

_life, 2016, doi:10.3390/life6010010_

Round 1

Reviewer 1 Report

The manuscript by H.S. Bernhardt presents a very plausible and interesting hypothesis on the early evolution of tRNAs, while making very logical arguments as to the origins of class II tRNAs. Importantly, the article presents a clear and stepwise explanation for the origin of such tRNAs and why certain amino acids group together in the genetic code table. Overall I think this is an excellent article, which is also thought provoking. The arguments are also presented in a manner were other proposals along the same lines are politely discussed and carefully considered. I only have very few suggestions that may improve clarity.

1. From the beginning, I believe it should be emphasize what period of tRNA evolution we are discussing here; I.e. the time period before protein synthetases.

I assume this is all thought to happen when we are still stuck in the RNA world. Please correct me if I am wrong.

2. Two mechanisms for the enlargement of the V-arm are suggested, the second as previously proposed could be the remnant of an intron. To avoid confusion, it should be specified that we are dealing with self-splicing introns (typical of bacterial introns) and not the more common tRNA introns found in Eukarya and Archaea, which are removed by protein enzymes. Clearly, only the self-splicing introns make sense here. In passing, I would have liked to hear how the author feels about the possibility of half tRNAs being generated through duplication and then creation of the long V-arm as the result of base-pairing and subsequent ligation; something similar to the modern day half tRNA of Nanoarchaea.

3. On page 5, when talking about the potential promiscuity of amino acylating ribozymes towards mis-acylation, it may be worth mentioning that even extant protein aaRSs are rather promiscuous; may requiring some type of editing domain.

4. Lastly, early in the discussion of glycine the pioneer work by Miller and, independently, Oro should be cited. Given that indeed glycine is one of the “easiest” amino acids to obtain in relatively large amounts under prebiotic conditions. I believe this is important because it gives further credence to the idea that glycine may have indeed been the first aminoacid used in “coded” peptide synthesis. 

Author Response

I thank the reviewer for their kind comments. In regards to the specific points raised:

I have added a paragraph at the end of the introduction to emphasize that my hypothesis addresses a period during the RNA world prior to the evolution of coded protein synthesis and complex peptides/proteins.

I have spelled out that the hypothesis deals with self-splicing introns.

I have included a sentence with references concerning the promiscuity of extant AARSs even in the presence of an editing domain.

I have included a sentence and references to Miller and Oro's amino acid synthesis in simulated prebiotic conditions near the beginning of the introduction.

Reviewer 2 Report

In this manuscript, the author extends his previous consideration of the early evolution of tRNA species. The manuscript is clearly written and includes several novel thoughts about the appearance, significance, and participation of the long variable loop present in Class II tRNAs (especially Fig. 3). As the long variable loops remain a stubborn and unresolved puzzle, the author’s thoughts may be a useful contribution to the literature on the development of the genetic code. Major revision should be made to address the following serious points:

1.     The chief logical difficulty with the paper is the assumption that the earliest transfer RNAs were well-defined genetic species, capable of evolution through point mutation and selection. This point of view is both facile and vulnerable to the criticism that the earliest kind of genetic coding very likely arose not from well-defined species, but from semi-random quasi-species. This assumption has always been questionable (1,2). It is clearly hard to sustain in light of recent theoretical (3,4) and experimental (5-7) work on the specificity of ancestral aminoacyl-tRNA synthetases. The manuscript does not address that vulnerability. As a result, the work remains little more than an interesting set of observations and attendant speculations that appear to be neither falsifiable nor otherwise experimentally testable.

2.     An especially notable corollary problem is that the review of literature on the evolution of the genetic code is quite limited. The evolutionary appearance of tRNAs must in some way be connected to that of the aminoacyl-tRNA synthetases. In particular, the treatment of the appearance of amino acids in the genetic code assumes that Class II amino acids appeared first, whereas the evidence that the original aminoacyl-tRNA synthetases appeared in pairs coded by opposite strands of the same gene is now quite substantial (8).

References

1.         Woese, C. (1969) Models for the Evolution of Codon Assignments. J. Mol. Biol. 43, 235-240

2.         Woese, C. R. (1965) On the Origin of the Genetic Code. Proc. Nat. Acad. Sci. USA 54, 1546-1552

3.         Wills, P. R. (2001) Autocatalysis, information, and coding. BioSystems 50, 49-57

4.         Nieselt-Struhe, K., and Wills, P. R. (1997) The Emergence of Genetic Coding in Physical Systems. J. theor. Biol. 187, 1–14

5.         Carter, C. W., Jr. (2015) What RNA World? Why a Peptide/RNA Partnership Merits Renewed Experimental Attention. Life 5, 294-320

6.         Weinreb, V., Li, L., Chandrasekaran, S. N., Koehl, P., Delarue, M., and Carter, C. W., Jr (2014) Enhanced Amino Acid Selection in Fully-Evolved Tryptophanyl-tRNA Synthetase, Relative to its Urzyme, Requires Domain Movement Sensed by the D1 Switch, a Remote, Dynamic Packing Motif J Biol Chem 289, 4367-4376

7.         Carter, C. W. J., Li, L., Weinreb, V., Collier, M., Gonzales-Rivera, K., Jimenez-Rodriguez, M., Erdogan, O., and Chandrasekharan, S. N. (2014) The Rodin-Ohno Hypothesis That Two Enzyme Superfamilies Descended from One Ancestral Gene:  An Unlikely Scenario for the Origins of Translation That Will Not Be Dismissed. Biology Direct 9, 11

8.         Martinez, L., Jimenez-Rodriguez, M., Gonzalez-Rivera, K., Williams, T., Li, L., Weinreb, V., Niranj Chandrasekaran, S., Collier, M., Ambroggio, X., Kuhlman, B., Erdogan, O., and Carter, C. W. J. (2015) Functional Class I and II Amino Acid Activating Enzymes Can Be Coded by Opposite Strands of the Same Gene. J. Biol. Chem. 290, 19710–19725

Author Response

After reasonably extensive reading of the papers referred to by this reviewer as well as a number of others by Carter, Rodin, Rodin and Ohno, I have made the decision not to address directly the issues raised by the reviewer. This was partly because of the formidable scope of these papers, which includes the evolution of tRNA and protein AARSs, as well as the relationship between tRNAs and amino acids in terms of hydropathy and size. Also, the paragraph I had written (which included six of the eight references provided by this reviewer), sat rather awkwardly alongside the rest of the MS.

Most importantly, I have come to the conclusion that this body of work is not directly relevant to the hypothesis presented here. Because of this (and also in response to a request from reviewer 1 for an explanation to this effect), I have sought to explain in a paragraph at the end of the introduction that my hypothesis adresses the RNA world, prior to the evolution of coded peptide/protein synthesis and complex protein enzymes. In taking this approach, I follow the lead of Professor Paul Schimmel who wrote (in his published review of Carter et al. 2014 Biology Direct 9:11) that in his opinion the work of Carter and the RNA world hypothesis can be harmonized "in a straightforward way", by assuming that the AARS 'urzymes' would have evolved following the evolution of ribozymic AARSs in an RNA world. And Prof Schimmel is not alone in this idea: Carter's collaborators Sergei and Andrei Rodin - who originally proposed a number of the fundamental ideas that Carter's theories are based on - had earlier written a paper entitled "Origin of the genetic code: first aminoacyl-tRNA synthetases could replace isofunctional ribozymes when only the second base of codons was established' (DNA and Cell Biology 2006, 25:365-375). In light of the above, I would hope that the reviewer and I could agree to disagree on some of the issues the reviewer has raised.

The above notwithstanding, I have included a reference to the 2008 publication by Rodin and Rodin in the journal Heredity (vol. 100:341-355; [ref 33] in the revised MS), as the hypothesis that tRNAs arose in complementary pairs with complementary anticodons offers a possible explanation for the complementary anticodons of the two types of serine tRNA I discuss in the manuscript.

Reviewer 3 Report

This is a well written hypothesis paper. I have only one content related doubt. The author argues about the function of Gly containing peptides. In my view, the first function of aminoacylation was to add functionality to the acceptor RNA of the amino acid. Hence transfer of Gly (probably the most abundant amino acid) would have been an easy way to add an amino group to RNA and provide a positive charge, which is not available in the chemical repertoire of RNA. Whether Gly had a specific role in the first peptides, I doubt, because of its characteristic lack of any functionality after being incorporated into a polypeptide chain.

Author Response

I thank the reviewer for their kind words. I have included a sentence in the final paragraph of the discussion (citing 'an anonymous reviewer') to the effect that an initial function of RNA aminoacylation may have been the added functionality of a protonated amine group to the chemical repertoire of RNA. 

Round 2

Reviewer 2 Report

I recommended publication of this manuscript in my first review because it presents an interesting perspective on the presence of the long variable loops in several tRNAs.

I cannot make a similar recommendation based on the author’s arrogant, stubborn, and totally inchoate response to peer review. LIFE can do as it sees fit with this manuscript. If it is published, the following comments should be placed alongside my previous remarks.

After reasonably extensive reading of the papers referred to by this reviewer as well as a number of others by Carter, Rodin, Rodin and Ohno, I have made the decision not to address directly the issues raised by the reviewer. ... the pararagraph I had written (which included six of the eight references provided by this reviewer), sat rather awkwardly alongside the rest of the MS.

This statement fully validates the original criticism. The author cannot write a response that doesn’t “sit awkwardly alongside the rest of the MS” because he cannot reconcile his ideas with relevant experimental data. The fact that the author cannot even cite those data proves the original point that the literature citations were both inadequate and misleading, and that although his observations may eventually be relevant, his hypothesis arising from them is difficult to reconcile with experiments.

Most importantly, I have come to the conclusion that this body of work is not directly relevant to the hypothesis presented here.

The author is unwilling even to consider that the RNA World hypothesis may be self-evidently preposterous.

And Prof Schimmel is not alone in this idea: Carter's collaborators Sergei and Andrei Rodin – who originally proposed a number of the fundamental ideas that Carter's theories are based on –

This statement is a curious, egregious misrepresentation. Although Carter’s papers discuss novel ideas based on a wide range of data, the papers themselves are in fact not theories, but represent perhaps the most coherent and relevant experimental data in a field in which most previous work had been exclusively theoretical.

Author Response

I was extremely taken aback by this review.

As the reviewer states, in his/her first report, s/he recommended publication; the reports of reviewers 1 and 3 were complimentary of the manuscript and – presumably – also favored publication (albeit with some fairly minor changes/additions):

Reviewer 3: This is a well written hypothesis paper...

Reviewer 1: The manuscript by H.S. Bernhardt presents a very plausible and interesting hypothesis on the early evolution of tRNAs, while making very logical arguments as to the origins of class II tRNAs. Importantly, the article presents a clear and stepwise explanation for the origin of such tRNAs and why certain amino acids group together in the genetic code table. Overall I think this is an excellent article, which is also thought provoking. The arguments are also presented in a manner were other proposals along the same lines are politely discussed and carefully considered...

Therefore, at this stage all three reviewers supported publication of the manuscript. In response, I revised the manuscript in accordance with their various suggestions except that I didn't feel able to address the voluminous literature of Charles Carter and colleagues apart from the inclusion of a single reference to the 2008 publication by Rodin and Rodin, ref. [33].

While I was and am still open to the possibility of including in the manuscript a paragraph referring to the work of Charles Carter (including citations of Peter Wills and Charles Carter's publications), I am disturbed by the angry – and somewhat personal – response of the reviewer to my revision, in which s/he calls me "arrogant" and "stubborn" and describes my response to his/her round 1 report an "inchoate response to peer review".

While I am open to the possibility that the RNA world–based view taken in my manuscript might be wrong or at least not not the whole story, the reviewer's argument that I need to "consider that the RNA World hypothesis may be self-evidently preposterous" is expressed in such highly emotive terms that it is hard not to conclude that s/he is passed beyond the realm of rational argument. Is the reviewer prepared to acknowledge the same, that the work of Carter et al. is "self-evidently preposterous"? Somehow I doubt it. I also do not think that the publications of Carter present the RNA world hypothesis as a (worthy) alternative to their ideas, and nor should they have to. When one reads these papers, one accepts that the author(s) are coming at the problem from their own perspective. On another note, I have no problem with Carter's experimental results – rather, it is the conclusions that he draws from them and the overarching theory/theories he argues they support that I have some difficulty with.

I freely acknowledge that there are alternative points of view to the RNA world hypothesis, but perhaps a more constructive response on the behalf of the reviewer would be to publish a response to my manuscript, perhaps in this same journal.

In the end, if you as the editor decide that my manuscript does indeed need to address the work of Charles Carter and colleagues, I would be prepared to revise the manuscript accordingly; however, it needs to be realized that my comments regarding his ideas will not necessarily be favorable. I am also in the middle of writing a funding application that needs to be submitted in three weeks time and so will not be in a position to start work on a revision until the end of February. My revision may therefore not be ready for 6–8 weeks.